# Virtual Screening with Gnina 1.0

**DOI:** 10.3390/molecules26237369

**Published:** 2021-12-04

**Authors:** Jocelyn Sunseri, David Ryan Koes

**Affiliations:** Department of Computational and Systems Biology, University of Pittsburgh, Pittsburgh, PA 15260, USA; jss97@pitt.edu

**Keywords:** virtual screening, structure-based drug design, molecular docking, deep learning

## Abstract

Virtual screening—predicting which compounds within a specified compound library bind to a target molecule, typically a protein—is a fundamental task in the field of drug discovery. Doing virtual screening well provides tangible practical benefits, including reduced drug development costs, faster time to therapeutic viability, and fewer unforeseen side effects. As with most applied computational tasks, the algorithms currently used to perform virtual screening feature inherent tradeoffs between speed and accuracy. Furthermore, even theoretically rigorous, computationally intensive methods may fail to account for important effects relevant to whether a given compound will ultimately be usable as a drug. Here we investigate the virtual screening performance of the recently released Gnina molecular docking software, which uses deep convolutional networks to score protein-ligand structures. We find, on average, that Gnina outperforms conventional empirical scoring. The default scoring in Gnina outperforms the empirical AutoDock Vina scoring function on 89 of the 117 targets of the DUD-E and LIT-PCBA virtual screening benchmarks with a median 1% early enrichment factor that is more than twice that of Vina. However, we also find that issues of bias linger in these sets, even when not used directly to train models, and this bias obfuscates to what extent machine learning models are achieving their performance through a sophisticated interpretation of molecular interactions versus fitting to non-informative simplistic property distributions.

## 1. Introduction

Virtual screening poses this problem: given a target molecule and a set of compounds, rank the compounds so that all those that are active relative to the target are ranked ahead of those that are inactive. An in vitro screen is the source of ground truth for this binding classification problem, but there are at least four significant limitations associated with such screening: time and cost limit the number of screens that can be run; only compounds that physically exist can be screened this way; the screening process is not always accurate; and in vitro activity against a given target is necessary but not sufficient for identifying useful drugs (perhaps this is a separate problem from virtual or in vitro screening, but from a practical standpoint it would be desirable to exclude compounds with problematic properties from the beginning of a drug discovery campaign, and, in theory, a virtual screening method could penalize such compounds in a ranking). Thus virtual screening has attracted significant interest as a way of overcoming these limitations to identify strong drug candidates at reduced cost.

Virtual screening methods can be broadly classified as ligand-based or structure-based. Ligand-based methods rely on information about known active compounds and base their predictions on the similarity between compounds in the screening database and these known actives. No 3D structures are required, but at least one known active is. There are many possible similarity metrics, but regardless of which is used, identifying truly novel actives with this approach is unlikely. In contrast, structure-based approaches derive from a model of the interaction between a protein and ligand, facilitating identification of truly novel interactions between the two. A “scoring function” maps the input structure representing the relative location and orientation of the pair of molecules to a score representing the strength of their interaction [1]. Several different approaches have been applied to scoring function development, yielding four major classes. Force fields [2,3,4,5,6,7,8,9,10], empirical scoring functions [11,12,13,14,15,16,17], and knowledge-based functions (also referred to as statistical potentials) [18,19,20,21,22,23,24] are known collectively as “classical” scoring functions, distinguishing them from the newer machine learning (ML) scoring functions [24,25,26,27,28,29,30,31]. Briefly, force fields rely on physics-based terms mostly representing electrostatic interactions; empirical scoring functions may include counts of specific features as well as physics-inspired pairwise potentials; and knowledge-based statistical potentials calculate close contacts between molecules in structural databases and fit potentials biased toward structures that resemble this reference data. In comparison, modern ML scoring functions tend to impose fewer restrictions on the final functional form and attempt to learn the relevant features from the data and prediction task itself (for example, they may consist of a neural network that processes the structural input directly).

Since structure-based approaches rely on a representation of the binding mode defined between the protein and ligand structures, the first step in using them is often generating one or more plausible binding modes. A typical approach is to start from a protein structure and use a scoring function to identify favorably scored conformations and binding poses of all compounds of interest (i.e., “docking”) within a search space defined on the surface of the protein. That scoring function may differ from the scoring function that will be used to generate the final compound ranking for the virtual screen; a persistent problem in this domain has been difficulty in simultaneously optimizing scoring functions for accurate binding pose scoring and accurate compound ranking. This “pose prediction” task should be fundamental to structure-based approaches to virtual screening, since these approaches aim to use the physical interactions underlying binding to guide scoring. If molecular interactions are not represented accurately by a pose used for scoring, the scoring method will either be unable to accurately score the pose, or will accurately score the pose for reasons unrelated to molecular interactions—i.e., it devolves to a ligand-based approach. In practice, it has been found that for many ML scoring functions, accurate input poses are not essential for good performance at binding affinity prediction [32].

Well-designed benchmarks can be constructed to require more than simple descriptors derived solely from the ligand to achieve good virtual screening performance. Benchmarks that are not designed to account for this bias are susceptible to delivering “state-of-the-art” performance when used to train and evaluate ML scoring functions merely because they can be perfectly classified using descriptors so simple that classical scoring functions would never be so naive as to use them as the sole basis of a scoring model [33,34,35]. Such biased benchmarks may have limited utility for evaluating an existing scoring function—good performance on the benchmark could derive from either a uselessly simple or a sophisticated model, and the dataset’s bias means that if the goal of the benchmark is to predict a model’s ability to perform well on an unknown dataset, the benchmark may only provide information about the model’s prospective performance on another dataset with the same bias. These benchmarks may be of limited utility for training machine learning scoring functions that generalize to real-world tasks, since training on them may merely produce a model that recapitulates their biases. Thus while these biased benchmarks could have served as acceptable assessments of classical scoring functions, where the explicit design choices made by human researchers eschewed the achievement of perfect performance via exploitation of dataset bias, fitting modern machine learning scoring functions to them risks creating models that have been “taught to the test” and cannot be expected to generalize beyond it.

Once problems with an existing dataset are identified, the challenge of constructing an improved alternative remains; this problem, combined with the need to compare new scoring functions with existing published results for older scoring functions (which may have exclusively had access to benchmarks now deemed problematic), ensures the continued relevance of now disfavored benchmarks. Such is the case with DUD [36], DUD-E [37], and MUV [38], three virtual screening benchmarks that have been widely used to assess scoring functions in the literature.

More recent literature [33,34,35] has demonstrated that both MUV and DUD-E are biased and are likely to be unsuitable for training or even validating machine learning scoring functions. Sieg et al. [33] found that for DUD, DUD-E, and MUV, better-than-random (and in the case of DUD and DUD-E, perfect) AUCs could be obtained merely by fitting cross-validated models on exactly the simple chemical descriptors that the dataset developers had attempted to control for during dataset construction. For DUD-E, synergistic effects were associated with using multiple descriptors together; the authors note that this probably derives from the construction process, which matches each feature separately in its one-dimensional feature space, unlike MUV, which considers distances within the multidimensional feature space. Accordingly, the authors find that MUV does not afford synergistic performance when including additional features. Wallach and Heifets [34] explain that MUV considered only the difference between active-active and active-inactive distances, omitting a comparison of inactive-inactive distances; since class labels for machine learning models are arbitrary, the MUV approach may produce datasets where “actives” are not clumped but the “inactives” are, and a machine learning scoring function can in principle learn from the intraclass similarity of either class. Further, as Sieg et al. [33] point out, the MUV dataset was constructed for ligand-based similarity search, and therefore it is likely to be inappropriate for benchmarking machine learning methods due to inherent analogue bias. Finally, Chen et al. [35] note that there is high similarity among inactives across targets in DUD-E, biasing that benchmark even further.

In their paper describing the limitations of only considering distances relative to actives in the MUV dataset construction approach, Wallach and Heifets [34] propose Asymmetric Validation Embedding (AVE), an improved measure of bias that considers clumping among inactives and between examples from the same class used in the training and validation sets. They do not construct a new dataset using AVE, however; rather, Tran-Nguyen et al. [39] first reported a novel dataset, LIT-PCBA, that used AVE for unbiasing and was explicitly designed for training and validation of machine learning scoring functions. It consists of 15 target sets, with 9780 actives and 407,839 inactives (some duplicated across multiple targets) after initial filtering. These values were reduced to 7844 unique actives and 407,381 unique inactives after AVE unbiasing. Thirteen of these targets have more than one PDB template provided as a reference receptor structure. All compounds are taken from assay data and therefore all inactives have experimental support for inactivity. The authors also confirmed that the included actives were not too biased toward high affinity compounds (i.e., the actives have typical potencies found in HTS decks) and that they were diverse when compared with other actives included for a given target. For all included targets, an EF1%>2 was achievable by at least one of a fingerprint-based, shape-based, or structure-based approach prior to AVE unbiasing (no such threshold was imposed on minimum performance for inclusion after unbiasing). Unfortunately, the majority of the primary assays used by LIT-PCBA are cell-based phenotypic assays (see Appendix A) and so most actives are not validated against their putative target. In fact, in at least one LIT-PCBA target (MAPK1) there are actives that were experimentally determined to selectively inhibit an alternative target (EGFR) [40]. This implies that, for target-based approaches, LIT-PCBA has an unknown number of incorrectly labeled actives.

Here we do not attempt to address the challenging problem of constructing a truly unbiased virtual screening benchmark appropriate for training machine learning models. Instead, we evaluate the convolutional neural network (CNN) models of the recently released Gnina 1.0 molecular docking software [41] on the established DUD-E [37] and LIT-PCBA [39] benchmarks. These models were trained for affinity prediction and pose selection and were not directly trained for virtual screening performance. We find that, on average, Gnina outperforms classical empirical scoring on these benchmarks. However, despite training for different outcomes and having minimal overlap in the raw training and test set data, we still find an underlying historical ligand-only bias that obfuscates the predictive power of these models.

## 2. Methods

We evaluate the built-in CNN models of Gnina on established virtual screening benchmarks and compare to multiple alternative scoring approaches, including evaluating ligand-only models trained on simple chemical descriptors.

### 2.1. Models

The Gnina approach to applying machine learning to molecular modeling is based on using 3D grids derived from voxelizing a fixed-size box centered on a protein binding site [42,43,44,45]. Our previous virtual screening evaluations [42,43,46] used older, less validated model architectures and were explicitly trained for the virtual screening task, which resulted in fitting to benchmark bias [33,35].

Here we evaluate the latest model ensembles [45] available in Gnina, which are based on the two architectures, Default2018 and Dense, shown in Figure 1. CNN models are used to score and rank poses generated using the AutoDock Vina [17] scoring function and Monte Carlo search, as integrating CNN scoring earlier in the docking pipeline was not found to be beneficial and came with a significant computational cost [41]. Gnina 1.0 contains four pre-trained model ensembles using these two model architectures and different training sets. These model ensembles contain five models trained with five random seeds. The default model ensemble (“Default”) is constructed from individual models of these four ensembles to balance computational cost and predictive performance [41]. In addition to evaluating this default ensemble, we also show results for the General ensemble, which combines the simplest model, Default2018, with the smallest training set, redocked poses from the 2016 PDBbind General set, and the Dense ensemble, which combines the largest model with the largest training set, CrossDocked2020 [45]. The variations in architecture and training data allow us to compare the effects of these aspects of the CNN scoring functions on virtual screening performance, while the ensembles themselves are expected to improve average predictive accuracy by reducing the effects of bias from individual learners [47] and in theory allow us to approximate the uncertainty in our predictions [48,49].

Note that none of these models were trained to perform virtual screening. Their outputs do not classify an input as “active” or “inactive” directly, nor were they provided distinctly “active“ or “inactive” compounds as input examples (i.e., they were not trained on any virtual screening datasets). Instead, they were simultaneously trained to predict whether a given input is a binding mode (<2Å RMSD) and, if so, what its affinity would be; if a pose is not a binding mode, the affinity is instead optimized to be lower (in pK units) than the true binding affinity for that compound. Consequently, the models produce both a pose score and an affinity prediction; e.g., a weak binder with a correct pose should have a high pose score and a low predicted affinity.

### 2.2. Metrics

We primarily use the area under the receiver operating characteristic curve (AUC) and top 1% enrichment factor (EF1%) to assess performance. The AUC assesses the quality of the entire ranking of compounds, with a perfect ranking receiving a 1.0 and a random ranking 0.5. From a practical standpoint, the ability of a method to provide a set of compounds highly enriched for actives as its top ranked compounds is highly desirable for virtual screening. EF1% is the ratio of the percentage of actives in the top 1% of ranked compounds to the overall percentage of actives. Unfortunately, the best possible EF1% varies depending on the number of actives and inactives, making it difficult to compare performance across benchmarks. To address this, the normalized EF1% [50] (NEF1%) divides the EF1% by the best achievable EF1% so that 1.0 means that as many actives as possible are ranked in the top 1% and zero means that none are.

### 2.3. Benchmarks

We use DUD-E and the more recently published LIT-PCBA dataset to assess virtual screening performance. DUD-E is primarily used to facilitate comparisons with published work. LIT-PCBA is appealing due to its apparently principled construction and the fact that all actives and inactives were drawn directly from a single assay per target; even though we do not use the training and validation splits (as the built-in Gnina models were not trained this way) and instead assess our performance on the full dataset, it still features diverse and more typical (lower) potency actives, and topological similarity between actives and inactives. Each target that was included in the final LIT-PCBA benchmark could reach at least an EF1% of two using a fingerprint-based, shape-based, or structure-based method prior to AVE unbiasing, further evidence of its suitability as a benchmark. Despite these virtues, its reliance on primarily cell-based assays and lack of target validation for active compounds may limit the best achievable performance on this benchmark. Nonetheless, the distinctly different methods of construction of the two datasets makes for an interesting contrast when evaluating virtual screening approaches (e.g., see score distributions in Appendix A).

Neither evaluation dataset was used for training. Nearly all the ligands in these datasets lack an experimentally determined protein-ligand structure. Previous work [42] found that when training without known poses (i.e., using computer-generated putative poses) the learned models were effectively ligand-based. DUD-E’s known bias also suggests that it is unsuitable for model fitting, but that does not *necessarily* imply that it is useless for evaluating a model fit on other data, as we do here, since the model is not fit to DUD-E’s biases (at least not directly, but there is still some risk of exploiting bias to gain performance due to shared bias *between* datasets). Further, there is utility in comparing model performance when testing on an independent dataset versus performing cross-validation, since improved performance at classification on a dataset when training on a subset of it could be due to dataset-wide bias artificially enhancing performance (as appears to be the case with DUD-E, where similarity among inactives between targets constitutes test set leakage [35]).

### 2.4. Comparisons

Since most of the poses we use for training and scoring are generated with the smina [11] fork of AutoDock Vina [17], we take Vina as our empirical scoring function baseline. We also compare with Vinardo [51], a modified version of the Vina scoring function that aims to improve performance at pose prediction, binding affinity prediction, and virtual screening. We also include virtual screening results from two versions of RFScore (RFScore-VS [52], which was trained on DUD-E, and RFScore-4 [32], which was trained on the 2014 PDBBind refined set). These two random forest based scoring functions are an interesting contrast to our approach: RFScore-4 has similar training data to ours but is a different type of statistical model that was fit to predict binding affinity with a different training strategy and distinct features, while RFScore-VS was trained specifically for virtual screening.

We used docked poses we had previously generated (and used for rescoring [31]) for DUD-E, obtained with the default smina arguments –seed 0 –autobox_add 4 –num_modes 9 and a box defined by the crystal ligand associated with the DUD-E reference receptor. For LIT-PCBA we used –seed 0 –autobox_add 16 –num_modes 20. We used our CNN models, RFScore-VS, and RFScore-4 to rescore and rank these poses generated with the Vina scoring function. For Vinardo scoring, we generated new poses using Vinardo to generate a new set of poses (e.g., appending –scoring vinardo to the command-line) as, unlike the ML scoring functions, it was designed to be incorporated into the full docking pipeline. A method’s best predicted score for a (target, compound) pair was taken as its prediction except where noted otherwise. For DUD-E there is a single reference receptor per target, while LIT-PCBA typically provides more than one. In the case of multiple reference receptors, we docked into all provided receptors and took the maximum score over all of them.

Finally, we also establish baseline performance using a variety of statistical models fit to our training datasets with the simple chemical descriptors used in the construction of DUD-E and MUV as their input features. These include linear and nonlinear regression models (Lasso, K-nearest neighbors, Decision Tree, Random Forest, Gradient Boosted Tree, and Support Vector regressors) available through sklearn [53]. The associated descriptors are shown in Table 1.

## 3. Results

First we summarize virtual screening performance of the Gnina convolutional neural networks, initially comparing with Vina, Vinardo, RFScore-4, and RFScore-VS. We also assess pose prediction performance on the reference receptors provided with LIT-PCBA, which in 13 out of 15 cases involve multiple protein templates and therefore can be used to construct cross-docking tasks. Finally, we attempt to explain aspects of the observed performance, in particular taking inspiration from Sieg et al. [33] and establishing a baseline ML model fit to the “simple” chemical descriptors calculated for our training sets (Table 1). We can thereby compare our performance on the test sets to this baseline in order to assess the potential influence of shared dataset bias on performance.

### 3.1. Virtual Screening Performance

Virtual screening performance is shown in Table 2, with AUCs shown in Figure 2, NEF1% in Figure 3, and EF1% in Appendix A. Per-target confidence intervals are provided in Appendix A. We provide AUCs for comparison with other literature, but NEF1%, which assesses early enrichment, affords a better measure of virtual screening performance.

For all models, average performance according to either metric is better on DUD-E than on LIT-PCBA. In the case of RFScore-VS, which has the best performance on DUD-E (median AUC of 0.96) and the worst performance on LIT-PCBA (median AUC of 0.60), the performance discrepancy between the two benchmarks suggests that its performance on DUD-E is not an accurate representation of its generalization ability, likely due to the data biases discussed previously. RFScore-4 has virtual screening performance comparable to other methods tested (particularly Vina), despite not being trained with inactive examples, which have previously been suggested to be essential [52] for good virtual screening performance. Among the CNN models, the affinity score tends to provide better virtual screening performance than the pose score, and the Dense models generally perform best. In most cases, the significantly faster Default ensemble performs nearly as well as the Dense ensemble (median AUCs of 0.79 and 0.61 for Default versus 0.80 and 0.62 for Dense on DUD-E and LIT-PCBA respectively), affirming its selection as the default model in Gnina.

The LIT-PCBA paper reports EF1% for three baseline methods: fingerprints, ligand shape overlap, and Surflex-Dock (SD), a structure-based docking method. Each target that was included in the final LIT-PCBA benchmark could reach at least an EF1% of two by at least one of those three methods prior to AVE unbiasing. Interestingly, there is no clear correlation between our observed performance and the previously reported performance. For example, there are targets that were amenable to Surflex-Dock (OPRK1, ADRB2) that most of our structure-based approaches performed poorly on, and targets where only ligand-based approaches were reported to perform well (ESR, IDH1) where most of our models performed well (see Appendix A). This could be due to sampling differences between Surflex-Dock and Vina/Vinardo, but it could also be evidence of ligand-based shape or 2D descriptors being incorporated into the ML models.

The CNN predictions (particularly the affinity values) outperform other approaches. On LIT-PCBA, which was designed to more closely resemble true HTS experiments and on which none of the methods were directly trained, all the CNN models exhibit a larger average early enrichment than the other methods (although the improvement is not always statistically significant). Across the 102 DUD-E targets and 15 LIT-PCBA targets, there are only 24 targets where Vina has a statistically significant improvement in NEF1% performance relative to Default ensemble affinity scoring, but the Default is significantly better for 89 targets compared to Vina. Full per-target comparisons of all models with the Default ensemble with 95% confidence intervals are shown in Appendix A.

### 3.2. Pose Prediction Performance

Next we examine the CNN ensemble’s pose prediction performance on the templates provided with LIT-PCBA. When more than one template was provided, we cross-docked each crystal ligand into every available non-cognate structure and used each scoring function to rank the resulting poses. The CNN models were used to rescore Vina-generated poses, and all these were compared with Vinardo, which was derived from Vina but intended to improve its pose prediction performance. Such an improvement did not manifest on this benchmark, as shown by the average fraction of compounds with a “good” (≤2Å RMSD) pose sampled at ranks 1, 3, and 5 in Figure 4 (per-target results are shown in Appendix A). The CNN models improve on Vina’s pose ranking, whether using the output from the pose layer (which was trained to predict whether a given pose is a binding mode) or affinity layer (which was trained to predict binding affinity, in a manner that is pose-sensitive). Interestingly, there is no statistically significant correlation between model performance at pose prediction and virtual screening performance (Figure S18), although we note there are orders of magnitude fewer ligands available for pose prediction performance estimation than for virtual screening.

### 3.3. Understanding Performance

We would like to understand the mechanisms underlying virtual screening performance; we would especially like to examine whether our predictions are pose sensitive, whether trivial descriptors are the primary basis of model performance, and whether performance is predictable based on similarity to training data.

First, we check whether our virtual screening predictions are pose sensitive by comparing NEF1% when basing a compound’s prediction on its highest- versus its lowest-ranked pose. The assumption is that the lowest-ranked pose will be the lowest quality and lack realistic protein-ligand interactions. A model that performs well with low quality poses is likely using primarily ligand-only information and is ignoring protein-ligand interactions. Figure 5 shows this assessment for the Default ensemble. Other methods are shown in Appendix A. All methods exhibit some pose sensitivity, with the top-ranked pose generally exhibiting better performance and the bottom ranked pose often providing no enrichment, but there are also cases where non-random performance is achievable with even the lowest-ranked pose, and every model also has at least one task for which choosing the lowest-ranked pose outperforms the highest-ranked one. This suggests that pose information is being used but (1) it is not always correct and (2) it is likely not the sole basis of the prediction.

Next we investigated the set of “simple chemical descriptors” that are known to afford perfect performance on DUD-E when used to fit models [33]. Since none of the CNN models were fit to DUD-E (nor indeed to any virtual screening dataset), we might hope to have avoided fitting models that derive their performance from these descriptors. However, these descriptors are useful because of historical bias in the underlying datasets from which most benchmarks are drawn, so it is entirely possible for models fit to other datasets to have a bias with respect to these descriptors. In Francoeur et al. [45], motivated by this consideration, we assessed similar “Simple Descriptor” models for performance at binding affinity prediction on PDBbind and Pocketome test sets and found them to have better-than-random and, in some cases, competitive to state-of-the-art performance. Therefore it seems necessary to compare virtual screening performance to the baseline established by one or more reasonably trained models that use these simple descriptors.

To that end, we establish a performance baseline by fitting a variety of linear and nonlinear regression models (Lasso, K-nearest neighbors, Decision Tree, Random Forest, Gradient Boosted Tree, and Support Vector regressors) available through sklearn [53] to the ligand affinity data associated with the CNN models’ training sets (PDBbind 2016 and CrossDock2020). For features, we use the descriptors used in the construction of DUD-E, the descriptors used in the construction of MUV (see Table 1), or ECFP4 fingerprints as implemented in OpenBabel [54]. Hyperparameter optimization was performed for all models via cross-validation on the PDBbind-Refined 2016 set. We then evaluate how these “simple descriptor” models perform at virtual screening on our test sets.

In Figure 6, we take the maximum performance per-target across any of the simple descriptor models and compare it with the Default ensemble affinity score. Comparisons with additional models are provided in Appendix A. Note that since the best performing model for each target is selected, this is not intended to be a fair comparison, but instead suggest an upper-bound for how well a ligand-only model can perform given the ideal model class, descriptors, and training set for a target. The Default ensemble generally performs well, outperforming all simple descriptor models on 77 out of 117 targets, but it is worth noting that even when cross-training for different purposes (affinity prediction vs. virtual screening) and on different training sets, simple, ligand-only descriptors can often exhibit better than random performance. Early enrichment performance for different descriptors and training sets is shown in Figure 7. Simple descriptor model performance seems to be uncorrelated with the size of the training set and to depend primarily on the chosen descriptors, with the simplest descriptors (DUD-E) performing best and the most complex (ECFP4) worst.

Next we consider similarity between training and benchmark datasets. Figure 8 plots the early enrichment performance (NEF1%) of the Default ensemble on each target versus the similarity between actives in the benchmark and training set compounds (other models are shown in Appendix A). Similarities are computed using the Tanimoto coefficient of ECFP4 fingerprints. Only actives are considered since the training set does not include any inactive compounds. For each target active, the maximum similarity with any training set compound is computed and the average of these similarities is taken to represent the similarity of that target’s actives with the training set. There is a statistically significant correlation (Spearman ρ of 0.45 and 0.50 for affinity and pose scoring, respectively) between training similarity and early enrichment performance. However, there exists a moderate correlation even for the non-ML models (Spearman ρ of 0.21 and 0.34 for Vina and Vinardo, respectively, see Appendix A), suggesting that this trend is not entirely due to learned training set bias.

### 3.4. Score Adjustment

Finally, we consider two straightforward combinations of the pose and affinity score (more sophisticated methods [48,56,57] of consensus scoring are left for future investigation). Pose and affinity scores are combined either by taking the predicted affinity of the pose with the best pose score, or by multiplying the affinity and pose scores. As shown in Figure 9, simply multiplying scores results in a modest boost in virtual screening performance, although the difference in score distributions has minimal statistical significance, especially compared to affinity scores (*p* = 0.053). Nonetheless, this multiplication score is generated in Gnina outputs as the CNN_VS score, for easier ranking of hits.

## 4. Conclusions

Dataset bias is a serious obstacle to applying data-driven approaches to solve problems in drug discovery. Unless care has been taken to assess the bias of a dataset and lack of bias, accordingly, machine learning models fit to that dataset will learn its bias. Since many of the existing biases are historical, it is entirely possible to subsequently evaluate performance on a test dataset that shares similar biases and inaccurately report improvements in generalization when in fact the resulting model is worse at generalizing than the conventional scoring functions that predate it. The community is still developing appropriate datasets and evaluation methods to ensure that we can effectively leverage data without fitting to the artifactual patterns it contains.

A recent study of 14 machine learning scoring functions for virtual screening found that none of them outperformed classical scoring functions, except for RFScore-VS, which only performed well on DUD-E (the dataset to which it was fit) [58]. Here we have demonstrated that machine learning models fit for binding affinity prediction and pose selection, specifically the CNN models of the Gnina molecular docking package, can be used for virtual screening, and they outperform classical empirical scoring methods. Further, we show that, in most cases, these models significantly outperform models fit to the same training data using simple chemical descriptors. Although there remains substantial room for improvement, these results support the use of Gnina as an alternative to AutoDock Vina or smina when performing virtual screens.

## Figures and Tables

**Figure 1 molecules-26-07369-f001:**
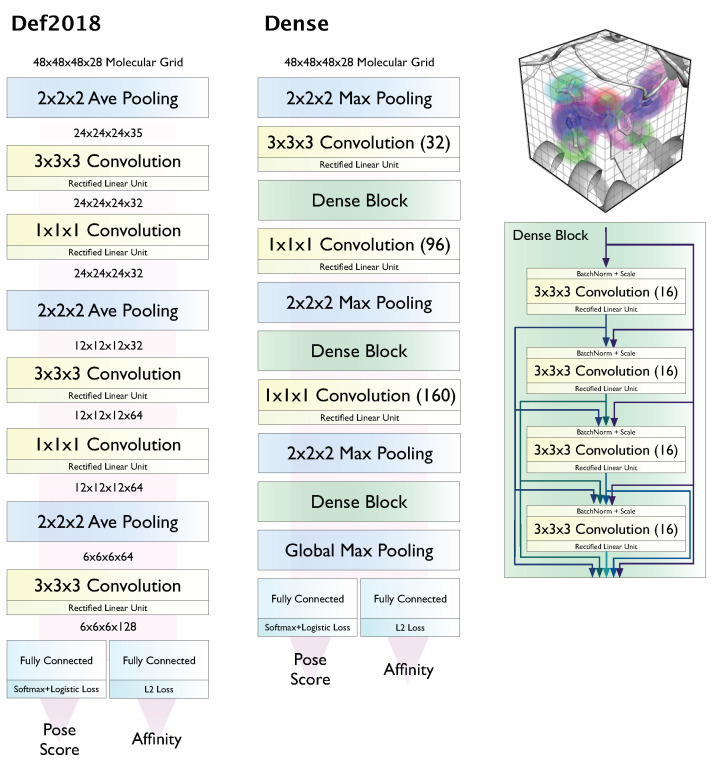
Voxelized grid-based CNN architectures evaluated in this work.

**Figure 2 molecules-26-07369-f002:**
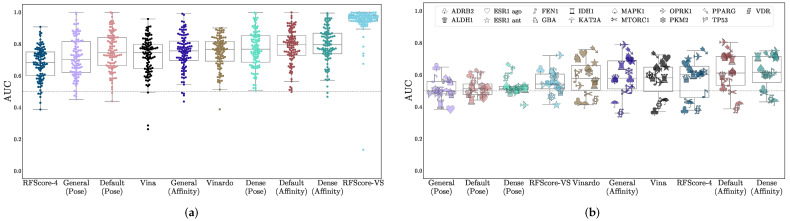
Assessment of virtual screening performance on (**a**) DUD-E and (**b**) LIT-PCBA using the AUC metric. The *x*-axis is sorted in order of increasing median performance. Each data point is the area under the curve of the ROC curve (AUC) of the method on a single target. LIT-PCBA targets are shown with distinctive individual markers.

**Figure 3 molecules-26-07369-f003:**
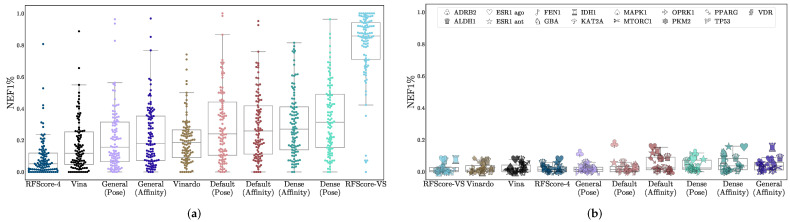
Assessment of virtual screening performance on (**a**) DUD-E and (**b**) LIT-PCBA using the NEF1% metric. The *x*-axis is sorted in order of increasing median performance. Each data point is the normalized 1% enrichment factor (NEF1%) of the method on a single target. LIT-PCBA targets are shown with distinctive individual markers. EF1% results are shown in Appendix A.

**Figure 4 molecules-26-07369-f004:**
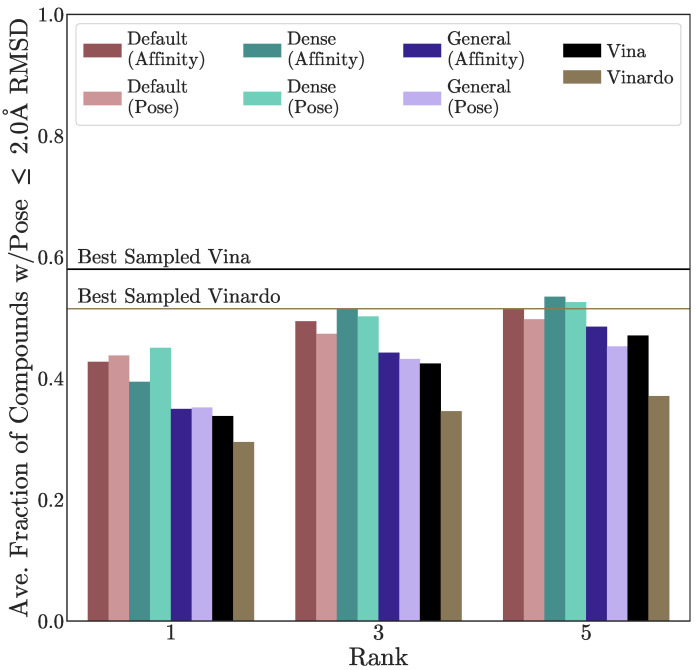
Assessment of cross-docking performance on LIT-PCBA structures. The percent of a target’s compounds with a good pose at ranks 1, 3, and 5, averaged across all thirteen targets in LIT-PCBA with more than one template available is shown. Labeled horizontal lines show the best performance possible with the poses sampled by Vina and Vinardo.

**Figure 5 molecules-26-07369-f005:**
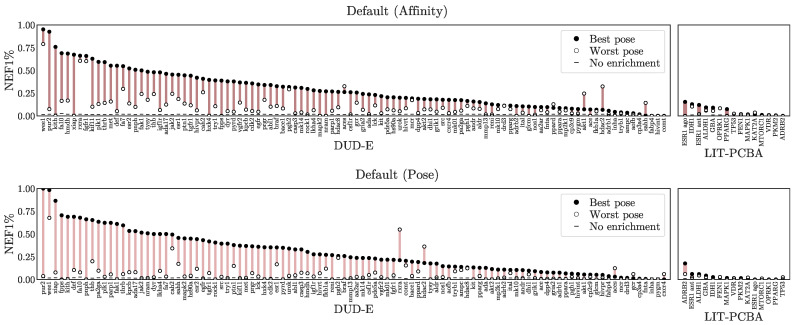
Pose sensitivity assessment. For each target the NEF1% achieved using the best scoring pose is contrasted with the NEF1% achieved using the worst scoring pose using the Default ensemble to score. Additional models and metrics are shown in Appendix A. The *x*-axes are sorted independently by the NEF1% of the best scoring pose.

**Figure 6 molecules-26-07369-f006:**
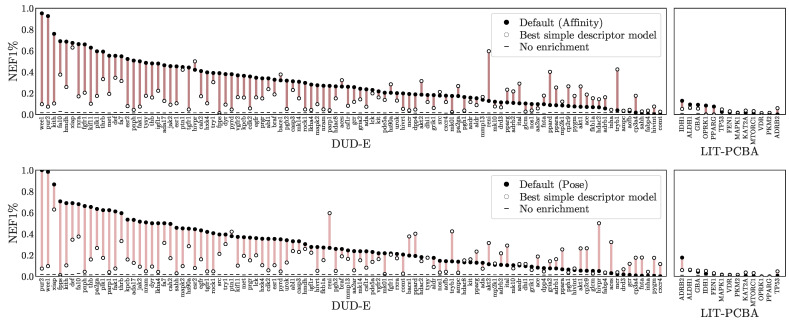
Comparison to simple descriptor models. The NEF1% for both the Default ensemble and the best performance achievable by a model fit to simple descriptors trained for affinity prediction on the PDBbind Refined set. Additional models and metrics are shown in Appendix A. The *x*-axes are sorted independently by the NEF1% of the best scoring pose.

**Figure 7 molecules-26-07369-f007:**
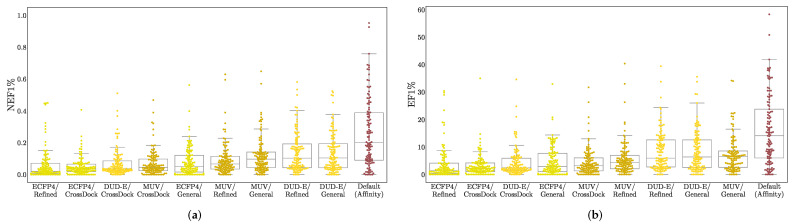
Early enrichment as measured by (**a**) NEF1% and (**b**) EF1% of best-performing ligand-only descriptor models when trained on different descriptors (ECFP4, MUV, DUDE-E—see Table 1) using training sets of different sizes and compositions (Refined and General from the PDBbind [55] and CrossDock2020 [45]). Each dot represents the best performing model for a DUD-E or LIT-PCBA target. Performance of the Default ensemble is provided for reference.

**Figure 8 molecules-26-07369-f008:**
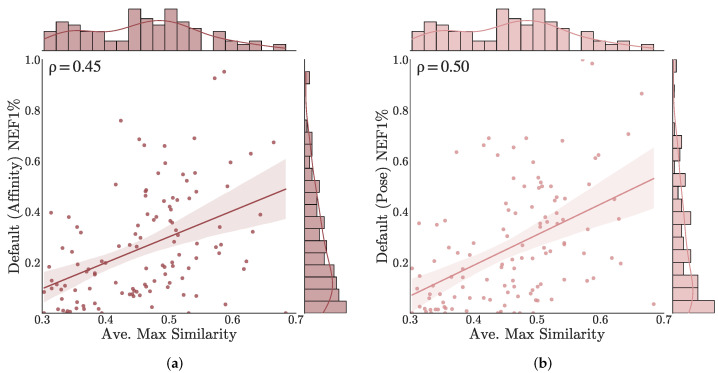
Correlation between similarity with training set and early enrichment performance for the Default ensemble (**a**) affinity and (**b**) pose scoring. For each benchmark, the average of the maximum similarity between active compounds and the PDBbind General set is computed using the Tanimoto coefficient of ECFP4 fingerprints.

**Figure 9 molecules-26-07369-f009:**
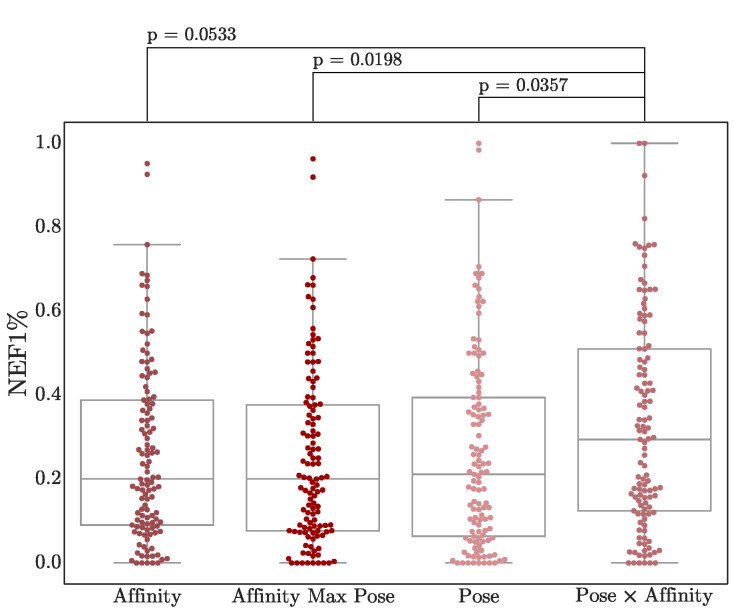
Early enrichment using score combinations. NEF1% performance of the Default ensemble on both DUD-E and LIT-PCBA targets is shown. The two-sided Mann-Whitney U rank test is used to compute *p*-values. Other metrics are shown in Appendix A.

**Table 1 molecules-26-07369-t001:** Descriptors used in the construction of DUD-E and MUV.

DUD-E	MUV
molecular weight	
number of hydrogen bond acceptors	number of hydrogen bond acceptors
number of hydrogen bond donors	number of hydrogen bond donors
number of rotatable bonds	
logP	logP
net charge	
	number of all atoms
	number of heavy atoms
	number of boron atoms
	number of bromine atoms
	number of carbon atoms
	number of chlorine atoms
	number of fluorine atoms
	number of iodine atoms
	number of nitrogen atoms
	number of oxygen atoms
	number of phosphorus atoms
	number of sulfur atoms
	number of chiral centers
	number of ring systems
6 features	17 features

**Table 2 molecules-26-07369-t002:** Median AUCs, NEF1% and EF1% values on DUD-E and LIT-PCBA. Mean values are provided in Appendix A. The best CNN model value for each column is shown in bold. Models whose distributions of per-benchmark metrics are not statistically dissimilar to the model in bold (as computed with a Mann-Whitney U rank test, *p*-value > 0.05) are shown in italic. RFScore-VS is the only model that was trained on DUD-E.

Model	DUD-E			LIT-PCBA		
	AUC	NEF1%	EF1%	AUC	NEF1%	EF1%
RFScore-4	0.683	0.0514	3.02	*0.6*	*0.013*	*1.28*
RFScore-VS	0.963	0.857	51.9	*0.542*	0.00733	0.733
Vina	0.745	0.118	7.05	*0.581*	*0.011*	1.1
Vinardo	0.764	0.187	11.4	*0.577*	0.0103	0.99
General (Affinity)	0.756	0.179	11.6	*0.579*	*0.037*	*2.06*
General (Pose)	0.702	0.156	10.3	0.498	*0.0147*	*1.3*
Dense (Affinity)	**0.795**	*0.27*	*17.7*	**0.616**	**0.037**	**2.58**
Dense (Pose)	*0.767*	**0.313**	**20.4**	0.514	*0.0238*	*1.81*
Default (Affinity)	*0.795*	*0.258*	*15.6*	*0.611*	*0.0238*	*1.88*
Default (Pose)	0.744	*0.241*	*15.8*	0.512	*0.0147*	*1.47*

## Data Availability

Docked poses and scores are available at http://bits.csb.pitt.edu/files/gninavs (accessed on 12 February 2021).

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
