# Peer review of "Virtual Screening with Gnina 1.0"

_molecules, 2021, doi:10.3390/molecules26237369_

Round 1

Reviewer 1 Report

In the present article Sunseri and Koes present the performances of their software, GNINA 1.0 in a virtual screening campaign, compared to the largely used docking program Autodock Vina and other scoring functions. The study completes the profile of the new software presented in a recent article on J Cheminf. To carry out the comparison, they used the well-known DUD-E dataset along with a more recent and less biased LIT-PCBA, highlighting the relevant role of this dataset composition in the evaluation of software, in particular for those exploiting machine-learning methods, such as GNINA.

The study is accurate, and many aspects are discussed in the final analysis.

The general presentation can be improved, some mistake is present along with the text (see below), and some parts could be simplified, such as the introduction that should be shortened.

Following, I reported some issues, but I suggest the authors deeply revise the manuscript. 

page 3 row 148: We find…, change in We found…

Figure 1 page 4, please rotate of 90° to facilitate the reading.

page 4 row 158: The GNINA approach to using…, change in …to use...

page 8 rows 300 and 301: perform well…performed well, change one of them.

page 9 row 339: …and is ignore protein…change in …and is ignoring protein…

Insert reference to the Figures in the text.

Author Response

We appreciate the reviewer's feedback and have corrected all grammatical mistakes and reoriented Figure 1.  All figures are referenced in the text.

Reviewer 2 Report

The manuscript, molecules-1463113, described the usage of GNINA 1.0 to virtual screening study. A thorough comparison has been done to get their conclusions that GNINA outperforms conventional empirical scoring. Virtual screening is a widely used technique in drug discovery, but, as the authors stated in the manuscript, there are truly some problems needed to be resolved. It is hard to say that which docking method is the best one up to now. GNINA is a good try to be a better program. In my opinion, this manuscript can be accepted for publication after a small revision.

  1. In Section 2.3, the authors stated that “we do not use the training and validation splits”. Why did the authors think that there is no need to split the data? Why is the validation unnecessary?
  2. I advise the authors read through the whole paper to avoid possible language mistakes, such as “that that” in Line 141.

Author Response

We appreciate the reviewer's feedback.

> In Section 2.3, the authors stated that “we do not use the training and validation splits”. Why did the authors think that there is no need to split the data? Why is the validation unnecessary?

We are evaluating the built-in models of GNINA, which were not trained using these training/test splits.  We have clarified this in the text.

I advise the authors read through the whole paper to avoid possible language mistakes, such as “that that” in Line 141.

Fixed.